# SARS-CoV-2 Omicron variations reveal mechanisms controlling cell entry dynamics and antibody neutralization

Enya Qing[1]*, Julisa Salgado[1], Alexandria Wilcox[2], Tom Gallagher[1]*

**1** Department of Microbiology and Immunology, Loyola University Chicago, Maywood, Illinois, United States of America, **2** McKelvey School of Engineering, Washington University in St. Louis, St. Louis, Missouri, United States of America

* eqing1@luc.edu (EQ); tgallag@luc.edu (TG)

**Data Availability Statement:** All data underlying the findings are fully available without restriction. All data are in the manuscript and/or supporting information files.

## Abstract

Severe Acute Respiratory Syndrome Coronavirus 2 (SARS-CoV-2) is adapting to continuous presence in humans. Transitions to endemic infection patterns are associated with changes in the spike (S) proteins that direct virus-cell entry. These changes generate antigenic drift and thereby allow virus maintenance in the face of prevalent human antiviral antibodies. These changes also fine tune virus-cell entry dynamics in ways that optimize transmission and infection into human cells. Focusing on the latter aspect, we evaluated the effects of several S protein substitutions on virus-cell membrane fusion, an essential final step in enveloped virus-cell entry. Membrane fusion is executed by integral-membrane "S2" domains, yet we found that substitutions in peripheral "S1" domains altered late-stage fusion dynamics, consistent with S1-S2 heterodimers cooperating throughout cell entry. A specific H655Y change in S1 stabilized a fusion-intermediate S protein conformation and thereby delayed membrane fusion. The H655Y change also sensitized viruses to neutralization by S2-targeting fusion-inhibitory peptides and stem-helix antibodies. The antibodies did not interfere with early fusion-activating steps; rather they targeted the latest stages of S2-directed membrane fusion in a novel neutralization mechanism. These findings demonstrate that single amino acid substitutions in the S proteins both reset viral entry—fusion kinetics and increase sensitivity to antibody neutralization. The results exemplify how selective forces driving SARS-CoV-2 fitness and antibody evasion operate together to shape SARS-CoV-2 evolution.

## Author summary

Most adaptive mutations endowing SARS-CoV-2 with increased human transmissibility and infectivity alter viral spike (S) protein structure and function. Orchestrated structural transitions in these multidomain S proteins mediate cell entry functions. Comparative analyses of adapted SARS-CoV-2 variants identified S domains conducting the rhythm of these transitions. The S1 adaptation H655Y slowed late virus-cell membrane fusion, revealing S1 as a timer for this essential and final cell entry event. The H655Y-directed

**Funding:** This research was supported by the National Institutes of Health (R21 AI 178391 to TG). The funders had no role in study design, data collection and interpretation, decision to submit the work for publication or preparation of the manuscript.

**Competing interests:** The authors have declared that no competing interests exist.

delay left viruses vulnerable to neutralizing antibodies that target transitional fusion intermediate S conformations. Our findings highlight how understandings of S protein structural dynamics illuminate antibody neutralization mechanisms and suggest that SARS-CoV-2 evolutionary pathways are shaped by competing selective pressures to optimize cell entry kinetics and evade neutralizing antibodies.

## Introduction

Severe Acute Respiratory Syndrome Coronavirus 2 (SARS-CoV-2) has been continuously adapting to human hosts since its emergence in late 2019. Most of these adaptations reside in the spike (S) proteins that direct virus-cell entry. D614G, the first S adaptation, increased virus stability and viral transmission, enabling subsequent adaptations [1–10]. Over time, variants of concern (VOCs) arose and replaced one another through increasingly effective transmissions. While the initial VOCs contained only 8~10 S adaptations, the latest Omicron strains had over 30 each, with all strains encompassing over 80 S adaptations in total [11–13]. Many S changes conferred antibody escape from neutralizing antibodies (nAbs), highlighting antigenic plasticity [14–20]. Some of these same nAb-resistance adaptations as well as other S changes impacted S protein functions at various stages of viral entry [17,18,21–24]. Yet there are other conserved substitutions with elusive functional ramifications. Identifying impacts of the changes can reveal underappreciated selective pressures driving alterations to the virus entry process.

CoV-cell entry proceeds through a series of well-calibrated steps. The S proteins are multi-domain homotrimers, with each monomer usually cleaved into S1 and S2. The peripheral S1 "cap" domains (Fig 1A) include the N-terminal S1A, S1B (the SARS-CoV-2 receptor binding domain), and domains C & D (S1CD). The transmembrane-anchored S2 "stalk" structures include alpha-helical heptad repeats (HR1 and HR2) that are required for membrane fusion [25–31]. S protein dynamics include S1B exposures to bind ACE2 receptors [27–29,32]. This is followed by three "mid-entry" events whose temporal sequence remains unclear [33–37]; these include S1 dissociation from S2, proteolytic cleavage within S2 at a site designated as S2', and embedment of a hydrophobic fusion peptide into host cell membranes, forming an extended intermediate conformation that bridges viral and host cell membranes. These steps are succeeded by a more well-characterized final event in which the extended intermediates refold into antiparallel HR1-HR2 helical bundles to bring the opposing membranes into the proximity needed for fusion. Our aims were to clarify the temporal sequence of mid-entry events such that entry dynamics and targets for antiviral agents might be further revealed.

The SARS-CoV-2 pandemic has incentivized research to understand how nAbs interfere with S protein functions. The findings in this area now make nAbs increasingly valuable tools for dissecting S protein dynamics. Broad nAb classes can be divided into those that prevent viruses from engaging host receptors [38–42] and those that prevent receptor-bound viruses from advancing through to the membrane fusion stage [43–48]. Members of the latter nAb class may have intriguing operating mechanisms that exert allosteric post-binding effects on S proteins [23,45,47,49]. One subclass of interest binds to S2 stem-helix epitopes (Fig 1A) that are well conserved across β-CoVs, but are mostly occluded in both the prefusion and the post-fusion S structures [45,47,50–58]. Plausibly, these stem-helix nAbs operate at mid-entry stages to neutralize infections. Our aims were to use these nAbs as probes in further defining the temporal sequence of transient CoV entry steps.

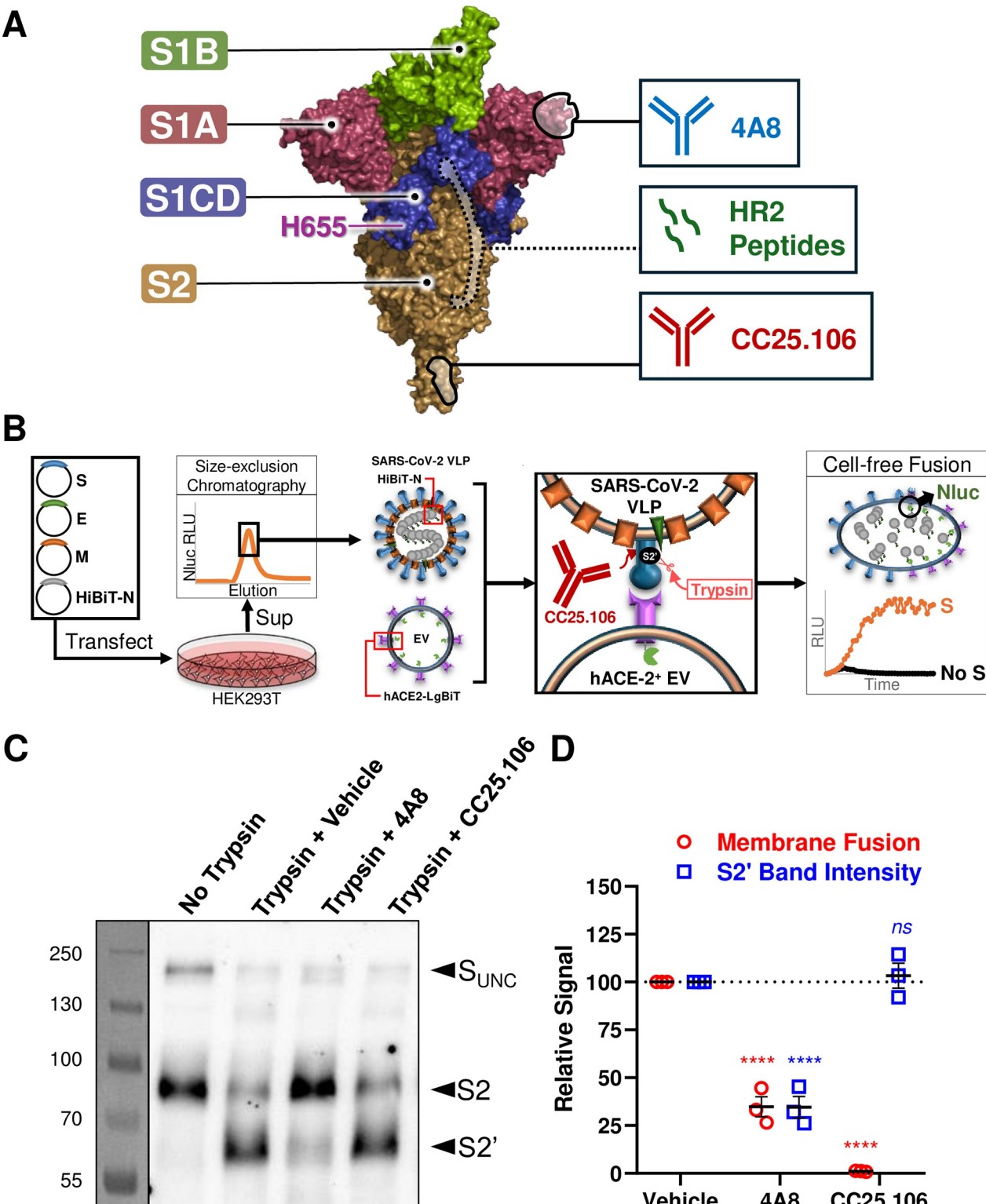

**Fig 1. SARS-CoV-2 S2 stem-helix antibodies neutralize S at a late stage of membrane fusion. A.** SARS-CoV-2 trimeric S prefusion structure (PDB: 7KRR). S1A (red), S1B (green), S1CD (blue), and S2 (brown) are labeled. Also labeled on a single S protomer are H655 and epitopes for the inhibitors used in this study, including: S1A antibody 4A8, HR2 peptides, and stem-helix antibody CC25.106. Of note, the HR2 epitope is largely hidden in the prefusion structure, and is labeled with dash outlines. **B.** The cell-free fusion system. VLPs were purified via size-exclusion chromatography from supernatant of cells transfected with SARS-CoV-2 structural gene expression plasmids. VLPs were mixed with hACE2+ EVs

and trypsin. Membrane fusion was measured as relative light units (RLUs) from nanoluciferase (Nluc) assembled via HiBiT (on the N protein) and LgBiT (on hACE2) complementation. **C and D.** HiBiT$^+$ VLPs were incubated with vehicle, 4A8, or CC25.106 (500 nM) at 37˚C. At 30 min, hACE2 EVs were added. At 60 min, vehicle or trypsin (1 ng/μl) was added. At 90 min, the samples were split for S fragment assessment via WB (panel C, S$_{UNC}$, S2, S2') or membrane fusion quantification as RLU readout relative to vehicle control (panel D). The experiment was repeated three times, and deviations from the vehicle control were analyzed by one-way ANOVA. Mean and standard error (SE) are graphed. ns, not significant; ****, p < 0.0001.

Utilizing our established cell-free SARS-CoV-2 membrane fusion assays, we found that a conserved Omicron substitution enhanced stem-helix nAb activity, but not by increasing nAb binding to prefusion S proteins. Evaluation of membrane fusion kinetics showed that the Omicron substitution stabilized extended intermediate conformations composed of complete S1-S2 heterodimers exposing epitopes that augment nAb neutralization potential. The findings bring out SARS-CoV-2 entry models that refine views of S protein dynamics and highlight potential conserved targets for antiviral agents.

## Results

### Stem-helix antibodies block a late step in SARS-CoV-2 spike-mediated membrane fusion

We used neutralizing antibodies to probe spike protein structural transitions taking place during virus entry. We focused on SARS-CoV-2 stem-helix-binding antibodies, which recognize a region in S2 (aa. 1145–1165, Fig 1A) immediately upstream of HR2 [28,29]. This region is marginally accessible to antibodies in both pre- and post-fusion S protein conformations, implying transient intermediate S structures exposing the antibody epitopes and becoming key neutralization targets [30,45,47,53,54,56,57]. To explore neutralization mechanisms, we utilized quantitative cell-free "entry" assays, in which SARS-CoV-2 virus-like particles (VLPs) interact with human (h) ACE2 extracellular vesicles (EVs) [10,23,24,59–62]. Complete virus entry is reflected by the VLP-EV coalescence that is measured by membrane fusion-dependent complementation of nanoluciferase reporter protein fragments (Fig 1B).

Stem-helix antibodies may block membrane fusion at a stage after receptor engagement [45,47,57]. As S protein cleavage at S2' substrate sites occurs after receptor engagement and before membrane fusion [10,23,27,63], we asked whether the stem-helix antibody CC25.106 [57] interfered with this fusion-activating proteolytic step. We have previously shown that the S1A antibody 4A8 (Fig 1A) neutralizes by impairing cleavage at S2' [23]. Indeed, 4A8 blocked viral membrane fusion in tight correlation with reduced levels of S2' cleavage fragments (Fig 1C and 1D). In contrast, stem-helix antibody CC25.106 did not block S2' proteolytic activation, yet potently inhibited membrane fusion (Fig 1C and 1D). These results indicate that S2 stem-helix antibodies block membrane fusion at a late stage after S2' proteolytic activation.

### Adaptive variations operate allosterically to increase stem-helix antibody neutralization

Many adaptive SARS-CoV-2 variants evade neutralization by ancestral antibodies; few if any do the opposite, increasing neutralization sensitivity. Expecting neutralization escape, we evaluated the neutralization of Omicron BA.1 by stem-helix antibodies. Intriguingly, antibody CC25.106 neutralized Omicron BA.1 more potently than ancestral D614G S (Fig 2A). Omicron BA.1 and D614G viruses have identical CC25.106 binding epitopes, therefore Omicron variations allosterically control stem-helix antibody neutralization potency.

To identify the responsible Omicron variations, we constructed domain-swapped D614G/ Omicron BA.1 chimeric VLPs (S1 Fig) and evaluated their relative neutralization sensitivities.

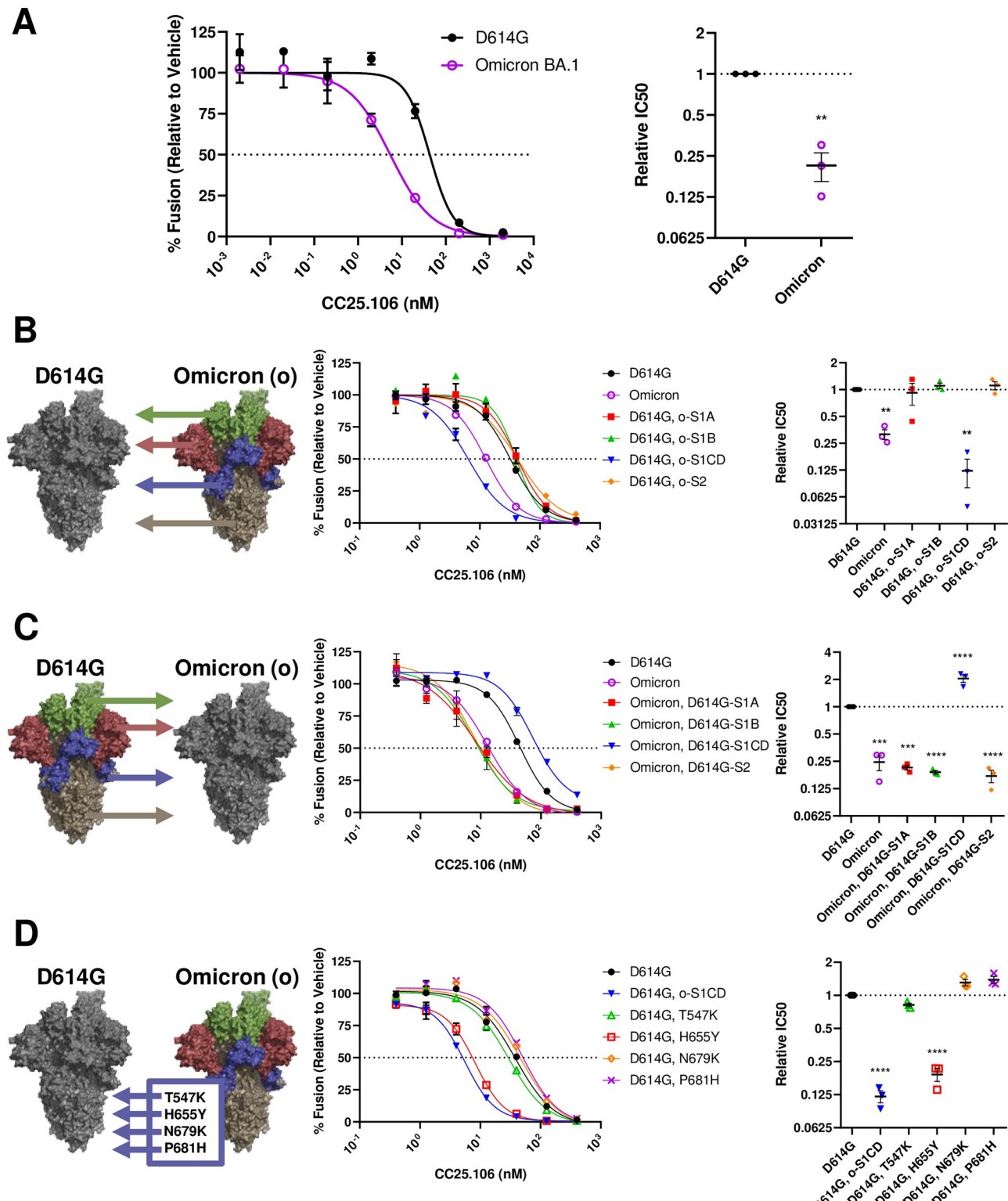

**Fig 2. H655Y sensitizes spike neutralization by S2 stem-helix antibodies. A.** The SARS-2 VLP cell-free fusion neutralization profile of stem-helix antibody CC25.106. Indicated concentrations of CC25.106 were incubated with VLPs bearing either D614G or Omicron BA.1 S for 30 min at 37°C before EVs, DrkBiT, substrate, and trypsin were added and RLU quantified. *Left*, dose-response curves between treatment and % fusion. *Right*, experiment on the left was repeated three times (N = 3) and the doses generating 50% inhibition (IC50) were plotted. Mean and SEM are depicted. Deviations from the reference value of 1.0 (D614G S) were analyzed by one-sample t tests. **, p < 0.01. **B, C, D.** CC25.106 neutralization profile of

various recombinant S proteins. *Left*, schematic indicating either mutations were taken from Omicron S domains to D614G S (**B**), from D614G S domains to Omicron S (**C**), or single S1CD mutations from Omicron S to D614G S (**D**). *Middle*, dose-response curves between treatment and % fusion. *Right*, experiment in the middle was repeated three times (N = 3) and the doses generating 50% inhibition (IC50) were plotted. Mean and SEM are depicted. Deviations from the reference value of 1.0 (D614G S) were analyzed by one-way ANOVA. **, p < 0.01; ***, p < 0.001; ****, p < 0.0001.

The changes sensitizing viruses to stem-helix antibody neutralization were within the Omicron BA.1 S1CD, far from the antibody binding epitope (Fig 1A). This was evident from the hyper-sensitivity of D614G chimeras harboring Omicron S1CD (Fig 2B; blue trendline) and the relative resistance of Omicron BA.1 chimeras with D614G S1CD (Fig 2C; blue trendline). To further elucidate the identity of the responsible mutation (s), we introduced each of the four Omicron BA.1 S1CD mutations (T547K, H655Y, N679K, P681H) into the D614G virus background. H655Y was the sole mutation responsible for the heightened neutralization sensitivity towards the stem-helix antibody CC25.106 (Fig 2D). Reinforcing these results, a reverting Y655H change in the Omicron BA.1 S background reduced the neutralization sensitivity (S2A Fig).

## The H655Y substitution does not increase stem-helix antibody binding to pre-fusion spikes

In native, pre-fusion SARS-CoV-2 S protein structures, the H655 residue is more than 5 nm away from the CC25.106 epitope [28,29], making H655Y-mediated exposure of the prefusion epitope appear unlikely (Fig 1A). Yet S protein structures in complex with CC25.106-like stem-helix antibodies show that the three S monomers in a trimer are never fully saturated by the antibodies [45,47,51,53,54,56]. Therefore, we considered the possibility that the H655Y change might change inter-monomer arrangements, allosterically exposing the CC25.106 epitope further to allow more antibody binding and enhanced neutralization.

Spike-antibody binding tests were established to test this possibility. In these assays, complete VLP-associated spikes were exposed to antibodies and then incubated to equilibrium. We chose this S protein presentation format because the VLP assembly and egress pathway mirrors that of authentic CoVs [64–68], making transmembrane S proteins on secreted VLPs faithful representations of infectious SARS-CoV-2. Alternative S ectodomains, or S proteins on cell or pseudovirus surfaces, may display unnatural S conformations [47,64,69,70].

VLPs were biotinylated and adhered to streptavidin magnetic beads (Fig 3A). Of note, the biotinylated VLPs were membrane fusion competent (Fig 3B), and were susceptible to proteolysis at the activating S2'cleavage sites (Fig 3C). The VLPs on magnetic beads also captured ACE2 EVs (Fig 3D). These findings are consistent with bead-based presentation of native prefusion S proteins.

We constructed antibodies with carboxy-terminal HiBiT tags and measured antibody binding to the immobilized VLPs with LgBiT complementation assays (Fig 3E). This approach was able to specifically and sensitively quantify antibody binding. Utilizing this system, we observed a low but significant binding of CC25.106 to native prefusion S (Fig 3F). The control S1A-binding 4A8 antibody bound to higher levels, likely because unlike CC25.106, 4A8 can bind to all three S protomers (Fig 1A) [43]. The presence of hACE2 EVs, which did engage and fuse with the VLPs, did not enhance CC25.106 antibody binding (Fig 3F).

VLPs with the H655Y substitution were included in the antibody binding assays. Here we found no H655Y-induced changes in CC25.106 binding (Fig 3G–3I). Proteolytic cleavage at the activating S2' site still took place in the absence of membrane fusion (Fig 3J). Therefore, we rejected the suggestion that the H655Y change increases antibody binding to prefusion spikes,

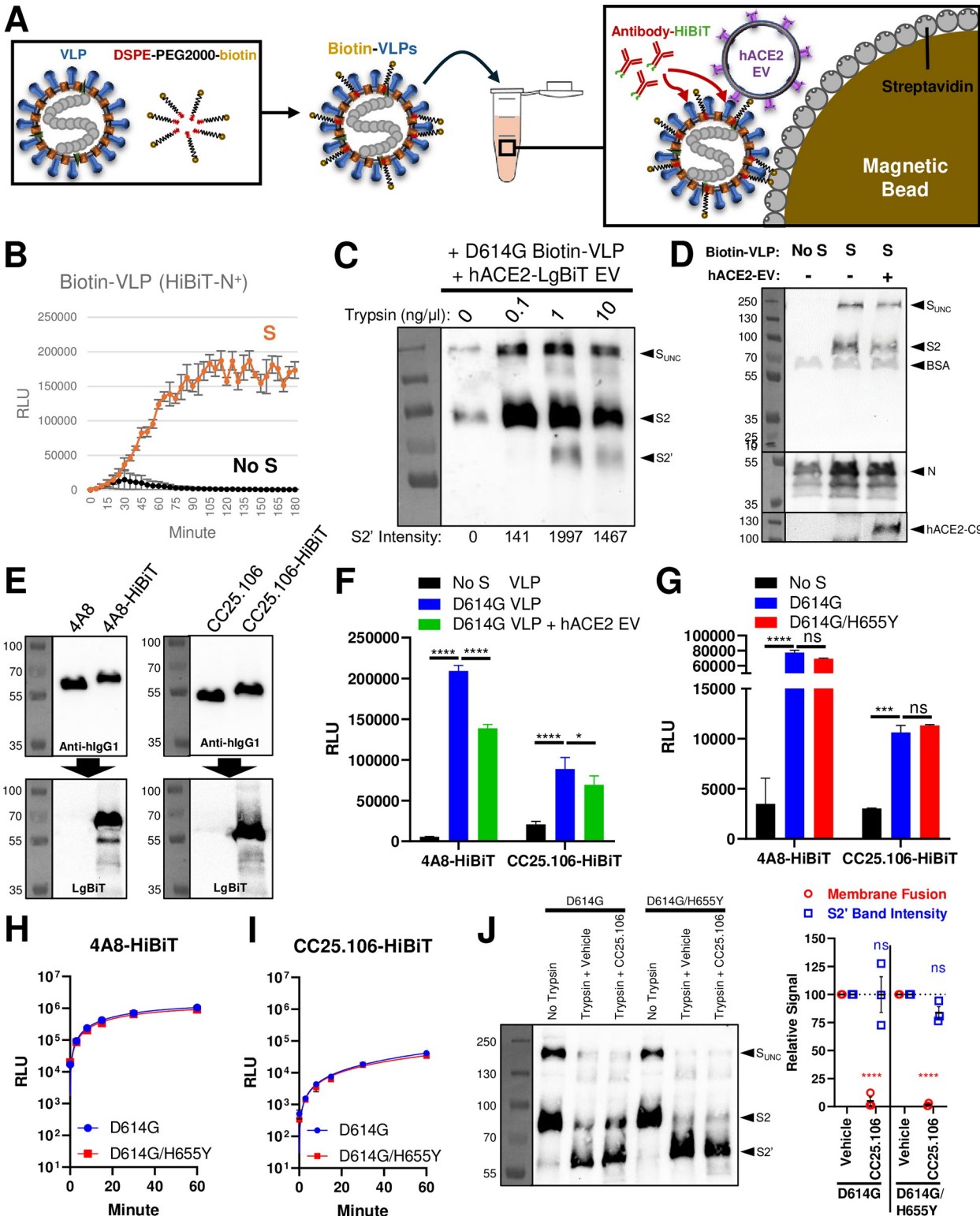

**Fig 3. H655Y does not affect stem-helix antibody binding to prefusion S. A.** Schematic for VLP-antibody binding assay. Purified VLPs (made with HiBiT-N or N) were biotinylated via coincubation with DSPE-PEG2000-biotin. The re-purified biotin-VLPs were mixed with HiBiT-tagged antibodies (4A8 or CC25.106) and hACE2 EVs. S-bound antibodies can be detected after magnetic pulldown of biotin-VLPs via streptavidin beads. **B.** The detection of S-mediated cell-free fusion of HiBiT-N[+] biotin-VLPs. **C.** WB of biotin-VLP trypsin S cleavage pattern in the presence of hACE2 EVs. S fragments (S$_{UNC}$, S2, S2') are labeled, and S2' band intensities are quantified. **D.** WB of streptavidin bead pulldown of VLP-EV complex. S

fragments ($S_{UNC}$, S2), BSA, N, and hACE2 are labeled. **E.** WB of HiBiT-tagged 4A8 or CC25.106 antibodies. The top blots used anti-hIgG1 to detect both tagged and untagged antibodies. The bottom blots were the same blots re-probed using LgBiT, which only detects HiBiT-tagged antibodies. **F.** RLU quantification of S-antibody binding. **G.** RLU quantification of S-antibody binding, comparing D614G S with or without H655Y. Statistical significance for panels F and G were analyzed via two-way ANOVA followed by Turkey test for multiple comparisons. Mean and STD are graphed. ns, not significant; *, p < 0.05; ***, p < 0.001; ****, p < 0.0001. **H and I.** The binding kinetics of 4A8 (**H**) or CC25.106 (**I**) to VLPs bearing D614G S with or without H655Y. Biotin-VLPs were pre-immobilized onto streptavidin beads before HiBiT-tagged antibodies were added at indicated times. Pulldown was carried out after 1 h, and antibody bound was measured via RLU. Results in panels B-I are representatives of two biological replicates. **J.** S2' cleavage assessment was done as described for Fig 1C and 1D. Upon completion, the samples were split for S fragment assessment via WB (*left*, $S_{UNC}$, S2, S2') or membrane fusion quantification as RLU readout relative to vehicle control (*right*). The experiment was repeated three times, and deviations from the vehicle control were analyzed by two-way ANOVA. Mean and standard error (SE) are graphed. ns, not significant; ****, p < 0.0001.

and instead argued that H655Y enhances stem-helix antibody neutralization by an undetermined effect on transitional intermediate S protein conformations.

## The H655Y substitution stabilizes S extended intermediate conformations

Transitional intermediate S protein conformations include "extended intermediates"; trimeric S2 helices connecting viruses to cells via transmembrane anchors and fusion peptides [33–37,62]. Current models propose that CoV S2 domains spring into extended intermediate states following proteolysis at S2' sites and then quickly collapse into helical bundles concomitant with membrane fusion [35–37,62].

HR2 peptides are potent membrane fusion inhibitors that bind HR1, a region exposed only in the extended intermediate conformation (Fig 1A, [62,71–80]), preventing the collapse process. We confirmed our previous observation that Omicron BA.1 S1CD adaptations elevated HR2 peptide sensitivity (Fig 4A, [24]). Remarkably, H655Y changes, but not the other Omicron BA.1 S1CD adaptations, were responsible for the enhanced HR2 peptide neutralization (Fig 4A). Conversely, Y655H reversion alone reduced Omicron BA.1 neutralization by HR2 peptides (S2B Fig). Of note, H655 residues are in an S1 domain that is not included in current models of S2-directed membrane fusion.

To determine whether the H655Y change heightened sensitivity to HR2 peptides by stabilizing the extended intermediate conformations, we added HR2 peptides at various intervals after mixing VLPs, EVs, and trypsin (Fig 4B). H655Y was the sole BA.1 S1CD change that prolonged HR2 peptide antiviral efficacy, extending the inhibitory time span relative to D614G ($\Delta T_{50}$) by 10–25 min (Fig 4C). Of note, the neutralization potencies of CC25.106 antibodies were similarly prolonged by H655Y (Fig 4D), indicating correlations between stem-helix antibody neutralization and extended intermediate durability. Soluble hACE2-Fc neutralizes by sterically interfering with virus-receptor binding; the efficacy of this antiviral reagent was expectedly unaffected by H655Y (Fig 4E). Together these results demonstrate that H655Y increases the dwell time of extended S conformations, in ways that prolong the presentation of HR2 peptide and stem-helix antibody binding sites.

We suspected that a durable extended S conformation would delay overall membrane fusion progression. This was validated using our cell-free fusion assays. Omicron BA.1 S1CD substitutions slowed D614G virus fusion kinetics ($\Delta T_{50}$ of 35 min), with H655Y being the only significant contributor (Fig 4F). Conversely, the Y655H reversion hastened Omicron BA.1 membrane fusion kinetics by 25 min (S2C Fig). There were associated caveats. The complete Omicron BA.1 VLPs fused membranes as rapidly as D614G (Fig 4G), consistent with Omicron BA.1 adaptations outside of S1CD opposing H655Y-directed fusion delays. Omicron BA.1 S1B domains bind hACE2 receptors more tightly than the ancestral D614G S1B [15,17,19,49,81,82], which could accelerate membrane fusion. Indeed, without the Omicron BA.1 S1B adaptations, fusion kinetics were slowed (Fig 4H), to a pace comparable to D614G S

 

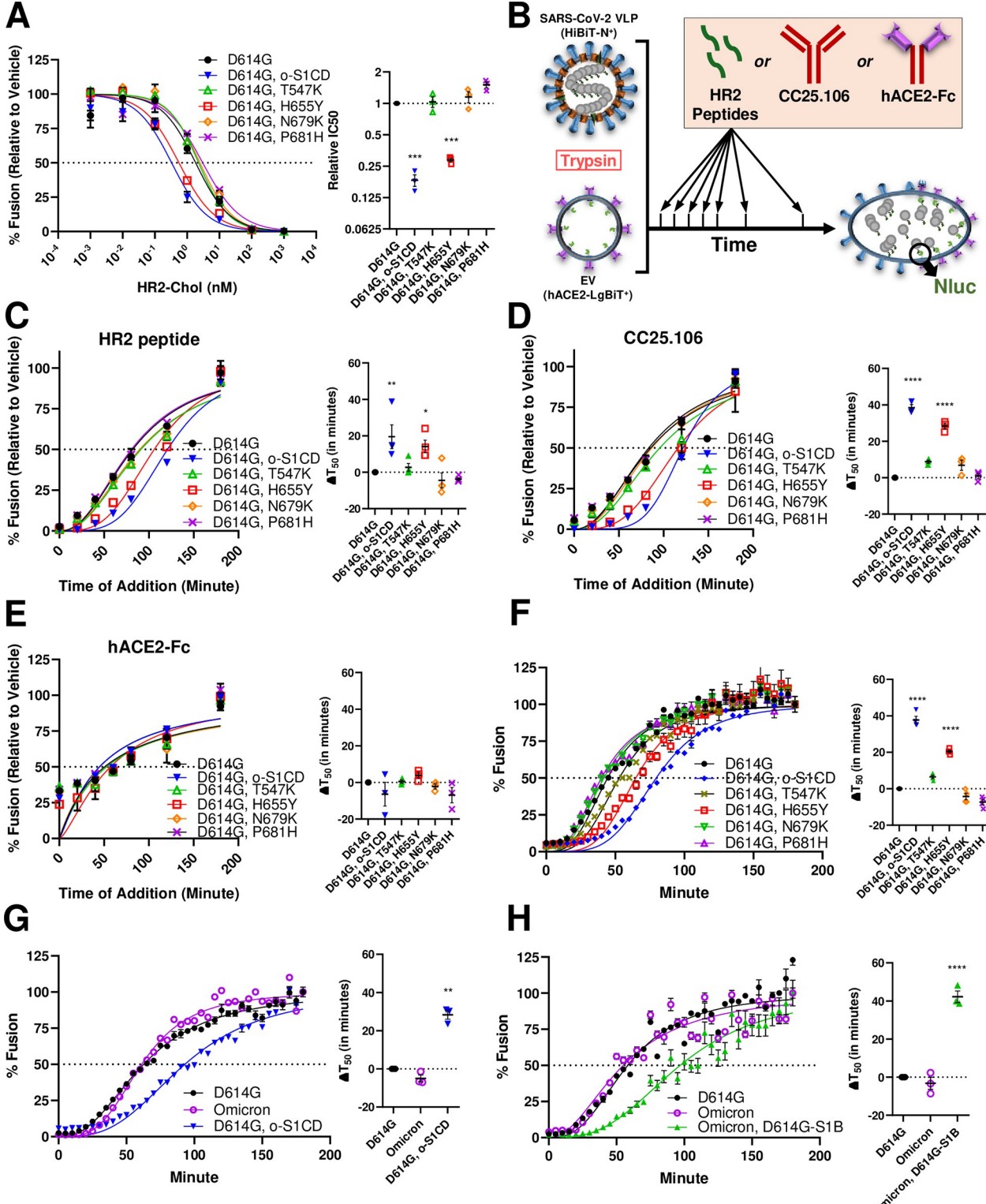

**Fig 4. H655Y slows down S-mediated membrane fusion by prolonging the extended intermediate S conformation. A.** The SARS-2 VLP cell-free fusion neutralization profile of HR2 peptides, the procedures as described for Fig 2A. *Left*, dose-response curves between treatment and % fusion. *Right*, experiment on the left was repeated three times (N = 3) and the doses generating 50% inhibition (IC50) were plotted. Mean and SEM are depicted. Deviations from the reference value of 1.0 (D614G S) were analyzed by one-way ANOVA. ***, p < 0.001. **B.** Schematic for fusion inhibitor time-of-addition experiments (**C-E**). VLPs were mixed with hACE2 EVs and trypsin and incubated at 37°C to allow membrane fusion. Then, at

various times, HR2 peptides (100 nM, panel **C**), CC25.106 (300 nM, panel **D**), or hACE2-Fc (63 nM, panel **E**) were added. RLU was read after 3 h at 37˚C. **C-E.** *Left*, dose-response curves between the indicated treatment and % fusion. *Right*, experiment on the left was repeated at least three times (N> = 3) and the dosing times generating 50% fusion ($T_{50}$) were plotted. Mean and SEM are depicted. Deviations from the reference value of 0 (D614G S) were analyzed by one-way ANOVA. *, $p < 0.05$; **, $p < 0.01$; ****, $p<0.0001$. **F-H.** Cell-free membrane fusion kinetics of VLPs bearing various recombinant S proteins. *Left*, fusion kinetics curves from a representative experiment, where RLU was read every 5 min up to 180 min. *Right*, experiment on the left was repeated at least three times (N> = 3) and the times generating 50% fusion ($T_{50}$) were plotted. Mean and SEM are depicted. Deviations from the reference value of 0 (D614G S) were analyzed by one-way ANOVA. **, $p < 0.01$; ****, $p<0.0001$.

with the Omicron BA.1 S1CD adaptations (Fig 4F and 4G). Together these results demonstrate that H655Y slows down membrane fusion by maintaining the extended S conformation. Conversely, Omicron BA.1 S1B adaptations compensate the overall fusion delay caused by H655Y. Evolution of SARS-CoV-2 is apparently subject to forces driving optimized fusion kinetics at specific entry steps.

## The H655Y substitution sensitized replicon particles to neutralization by stem-helix antibodies and HR2 peptides

Lastly, we sought to validate the observed H655Y characteristics in live-cell virus entry assays. To this end, we utilized a replicon-based transduction system that can be safely utilized in our BSL-2 laboratories [83,84]. This system replaces the SARS-CoV-2 S gene with co-expressing Gaussia luciferase (Gluc) and Green Fluorescent Protein (GFP) genes. Therefore, entry-competent virus particles can only be generated when a viral fusogen is supplied in trans (Fig 5A). Secreted particles with incorporated viral fusion proteins transduce single-cycle replicons, whose activities are recognized by Gluc and GFP (Fig 5B and 5C). Control tests showed that a SARS-CoV-2 S1B antibody prevented transductions directed by SARS-CoV-2 S but not by VSVG (Fig 5C).

We then generated replicon particles bearing D614G, Omicron BA.1, or D614G/H655Y S proteins. Compared to the ancestral D614G, Omicron and D614G/H655Y particles were preferentially neutralized by CC25.106 stem-helix antibodies (Fig 5D) and HR2 peptides (Fig 5E). These results concur with those from the cell-free fusion assays (Figs 2 and 4), with the reductionist cell-free assays providing less variability and higher resolution.

## Discussion

The past five years of SARS-CoV-2 evolution has now generated an assemblage of Omicron lineages competing for continuous transmission and infection in humans. In this Omicron collection there are more than 80 deletions and substitutions in the adaptable entry-facilitating SARS-CoV-2 S proteins. While many of these variations confer escape from pre-existing antibodies [14–20], others have more enigmatic effects on virus entry functions [17,18,21–24]. Entry requires extensive S protein conformational dynamics [35–37,62]. Aiming to add insights into orchestrated CoV entry, we hypothesized that the conductors are impacted by adaptive variations and we then went forward to determine whether substitutions in particular S domains exert kinetic controls. Specifically, we found that an Omicron substitution in S1, H655Y, recalibrated SARS-CoV-2 entry steps by stabilizing the extended intermediate S conformations that are poised for virus-cell membrane fusion (Fig 6A). This reset, in turn, sensitized SARS-CoV-2 to neutralization by an ancestral antibody that targets the exposed viral fusion machinery (Fig 6B). The findings demonstrate how understanding of S protein conformational dynamics brings insights on mechanisms of virus neutralization.

## A revised model of spike conformational dynamics during virus entry

SARS-CoV-2 entry begins with peripheral S1 domains engaging host cell receptors, with subsequent cleavage at S2' sites by host proteases [10,23,27,63]. The proteolysis enables extension

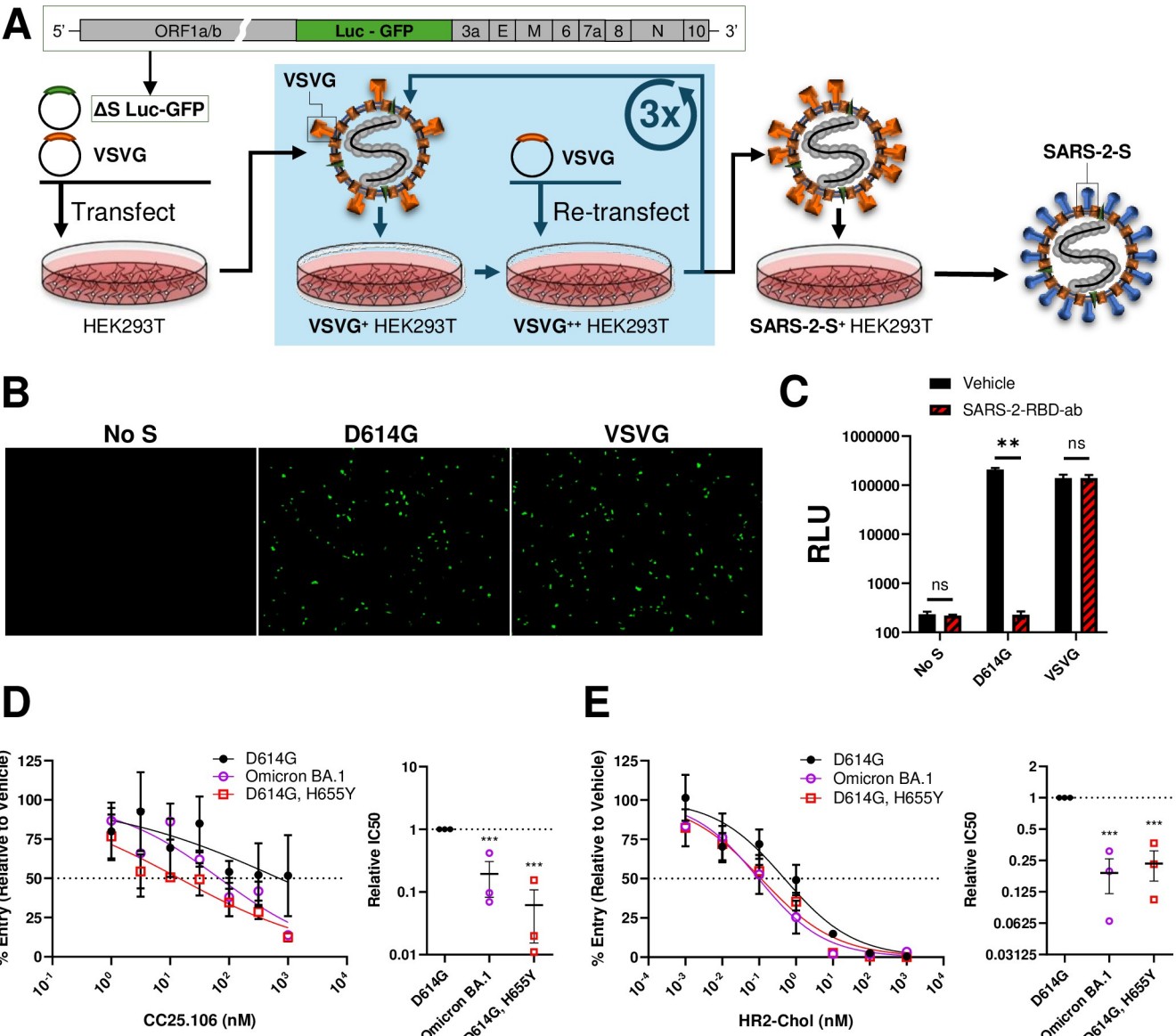

**Fig 5. H655Y sensitizes CoV replicon viral particle neutralization by S2 stem-helix antibodies and HR2 peptides. A.** Schematic of SARS-CoV-2 replicon virion generation. VSVG-pseudotyped virions were initially recovered from supernatant of HEK293T cells transfected with Bacmid ΔS Luc-GFP and VSVG expression plasmid. This supernatant then went through three rounds of enrichment in VSVG-transfected cells. The resulting particles were used to transduce CoV S-transfected HEK293T cells to generate single-round CoV virions. **B.** GFP fluorescent micrograph of Vero-E6-hACE2-hTMPRSS2 cells 24 h post inoculation with replicon virions bearing No S, D614G S, or VSVG. **C.** RLU readout of Vero-E6-hACE2-hTMPRSS2 cells 24 h post inoculation with replicon virions. Mean and STD are depicted. Statistical significances were analyzed by one-way ANOVA followed by Sidak test for multiple comparisons. n. s., not significant; **, p < 0.01. Experiments in panels **B** and **C** were repeated two times. **D and E.** Inhibitory profile of CC25.106 (**D**) or HR2 peptides (**E**) on SARS-CoV-2 replicon virion entry. Indicated concentrations of inhibitors were incubated with virions bearing either D614G, Omicron BA.1, or D614G/ H655Y S were incubated for 30 min at 37°C before inoculating onto Vero-E6-hACE2-hTMPRSS2 cells. Cells were rinsed after 2 h and incubated overnight, before RLU was read. *Left*, dose-response curves between treatment and % entry. *Right*, experiment on the left was repeated three times (N = 3) and the doses generating 50% inhibition (IC50) were plotted. Mean and SEM are depicted. Deviations from the reference value of 1.0 (D614G S) were analyzed by one-way ANOVA. ***, p < 0.001.

of integral-membrane S2 domains, in which several short helices and loops straighten into continuous helices that terminate at host membrane-embedded fusion peptides [30,31,62,85]. This transient extended intermediate trimer then collapses into the hyperstable antiparallel 6-helical bundles characteristic of class I fusion proteins [30,31,85–87], which draws virus and

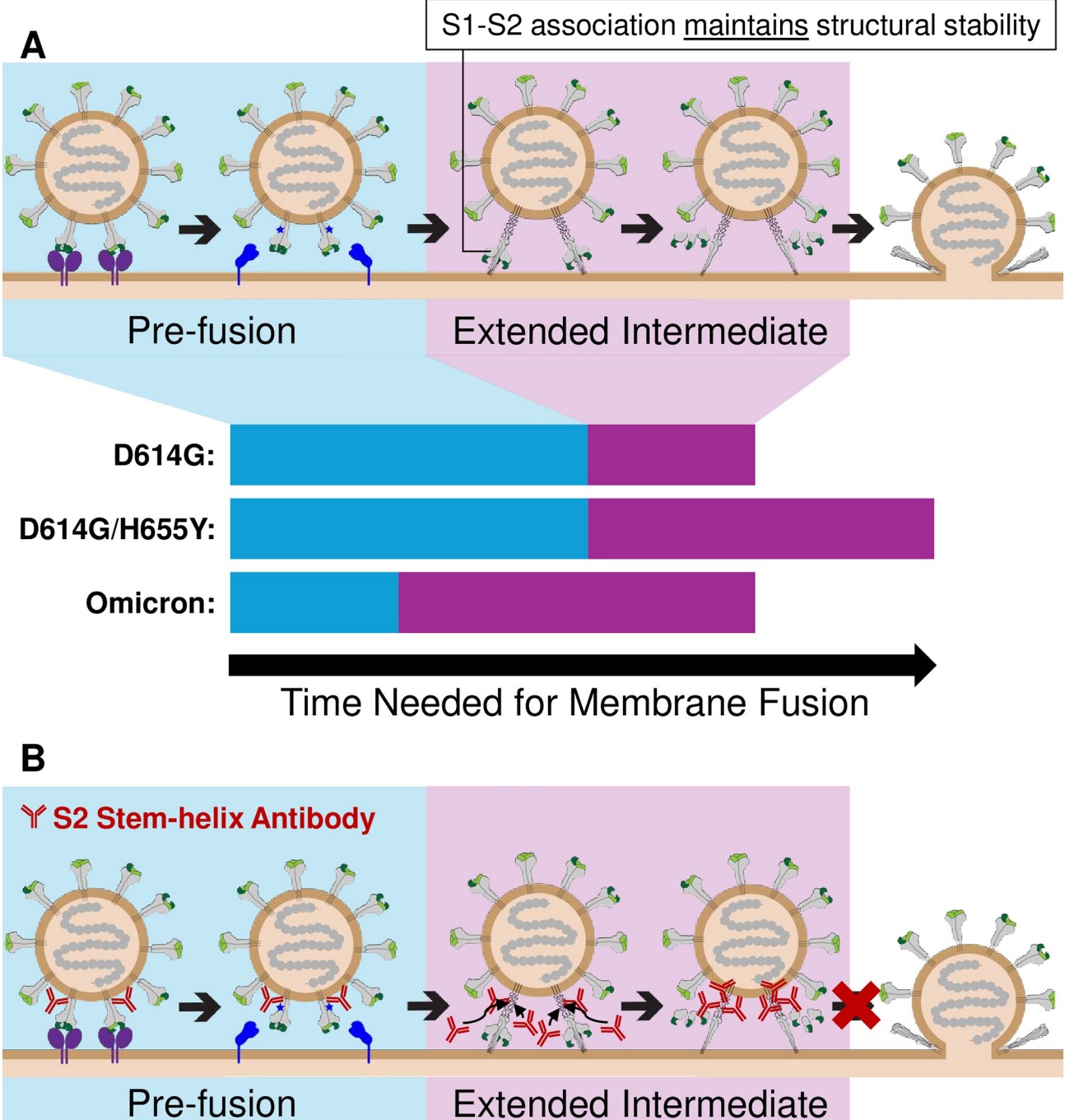

**Fig 6. H655Y functionality and the neutralization mechanism of S2 stem-helix antibodies. A.** Schematic for an updated depiction of CoV S-mediated membrane fusion progression. Instead of the assumed early dissociation of S1, S1 instead remains associated with S2 through S2' cleavage and the formation of the extended intermediate S conformation. The subsequently S1 dissociation allows the collapse of S2 from the extended intermediate into the ultrastable postfusion conformation. Pan-Omicron adaptation H655Y stabilizes S1-S2 interactions that prolongs the extended intermediate stage, lengthening the overall membrane fusion process. Omicron adaptations in the receptor-binding S1B shorten the pre-fusion stage, counteracting

H655Y's delay of membrane fusion overall. **B.** Schematic for a proposed relationship between stem-helix antibody S neutralization mechanism and the duration of the extended intermediate S stage. Pre-fusion S facilitates suboptimal levels of antibody binding due to steric hindrance. More extensive antibody binding is achieved with fully exposed antibody epitopes when S transitions to the extended intermediate conformation. Membrane fusion is blocked when sufficient levels of antibody prevent the full collapse of S2 into postfusion conformation. H655Y prolongs the extended S conformation, allowing more opportunities for stem-helix antibody binding, resulting in more effective S neutralization.

cell membranes together. It is generally assumed that S1 separation from S2 is a prerequisite for extended intermediate formation [33–37]. But this assumption cannot explain how H655Y, a substitution in an S1 domain, controls the timing of extended intermediate collapse into postfusion forms (Fig 4A and 4C).

For the H655Y substitution to increase the duration of extended intermediate states, S1 would need to remain associated with S2. We therefore propose an updated depiction of CoV entry, where S1 or S1 proteolytic fragments stay associated with S2 during extended intermediate formation (Fig 6A, top). With a trimer of S1-S2 heterodimers comprising the extended intermediates, the depiction proposes that eventual S1 dissociation enables the collapse of S2 into postfusion bundles that drives membrane fusion. In this model, H655Y indirectly or directly stabilizes S1-S2 interactions that prolongs the durability of the extended intermediate (Fig 6A, bottom; Fig 4A and 4C).

## Omicron recalibrates virus entry steps

The H655Y change, conserved in all Omicron sub-lineages [11–13], lengthened the total time needed for membrane fusion (Figs 4F and S2C). This reset in time-to-fusion may be powerfully selected in particular infection contexts. In cell cultures, the H655Y change shifts virus entry toward a "late" endosome-localized virus-cell fusion [22,88–90]. Conceivably this late entry arises because, with the fusion delayed by H655Y (Figs 4F and S2C), virus endocytosis precedes the fold-back of S extended intermediates; however it is worth noting that Omicron's endosomal entry prior to membrane fusion may be a cell culture-specific phenomenon [22,88,89,91,92], and may not take place *in vivo* [93–98]. Omicron lineages have an *in vivo* viral tropism for the ciliated cells of the nasal epithelia [92,94,97,99–102], and *ex vivo* nasal epithelial cell infections show no evidence of virus' preferential utilization of endosomal cathepsins [93–98]. Therefore, the utility of prolonged extended intermediates for *in vivo* infection may be explained by Omicron virus acquisition of unidentified entry factor(s) in human airways. Delayed virus-cell fusion may also allow viruses attaching at epithelial cilial apices enough time to traffic to cilial bases where viral genome delivery can more likely initiate productive infections [97].

Contextual selective forces may set time-to-fusion meters that are optimized for distinct infection environments. In support of this contention, we identified Omicron BA.1 S1B adaptations countering the H655Y-imposed fusion delays, resulting in an overall fusion kinetics comparable to the ancestral D614G S (Fig 4G and 4H). The S1B changes increase virus affinity for hACE2 receptors [15,17,19,49,81,82], and therefore we suggest that tight receptor engagement shortens the total time needed for viral entry. It appears that distinct entry steps can be separately programmed through adaptive evolution to set the time and place for final essential fusion events.

## Stem-helix antibodies relied on the extended intermediate S conformation for effective neutralization

Stem-helix antibodies receive considerable attention because they bind at low nanomolar affinities to epitopes that are well conserved across β-CoVs [45,47,50–57] and are powerfully

neutralizing *in vitro* and protective *in vivo* [45,50,52,54,55,57]. Neutralization by these antibodies were proposed to involve S protein conformations arising after virus-receptor binding [45,47,51,57]. We further narrowed the neutralization mechanism to post S2' cleavage (Fig 1C and 1D), and showed that the Omicron H655Y change increased stem-helix antibody neutralization (Fig 2D), without affecting antibody binding to prefusion S (Fig 3G–3I) or S2' activation (Fig 3J). Using timed addition of fusion inhibitors to track membrane fusion kinetics, stem-helix antibody neutralization was strongly correlated with stalled extended intermediates (Fig 4C and 4D).

Incorporating these observations, we propose the following neutralization mechanism for SARS-CoV-2 S2 stem-helix antibodies (Fig 6B). Initially, stem-helix antibodies bind to prefusion S proteins (Fig 3F–3I), with steric hindrances in the tightly packed S trimer partially burying antibody epitopes and preventing antibody saturation [45,47,51,53,54,56]. Antibodies bound at this stage do not impede S protein structural transitions through to proteolytic cleavage at S2' (Fig 1C and 1D). Subsequently, when S progresses to extended intermediate conformations, antibody epitopes become fully exposed, facilitating additional antibody binding. Bound antibodies then impose steric hindrances preventing S progression to postfusion conformations, as confirmed by a recent study [103]. This model is supported by the effects of the H655Y substitution, which have no effect on antibody binding to prefusion S proteins (Fig 3G–3I), but do prolong extended S conformations (Fig 4C), plausibly enabling more extensive antibody binding to this transient stage (Fig 4D), and result in an elevated neutralization profile (Fig 2D). We believe our findings will contribute to the development of broadly-neutralizing, high affinity stem-helix antibodies as effective antiviral agents [45,47,53,55,57], similar to those recently identified and characterized for measles therapy [87].

The pan-Omicron H655Y substitution rendered viruses hypersensitive to neutralization by stem-helix antibodies (Figs 2 and S2A), while nearly all previous virus neutralization assays have shown that viruses acquire mutations conferring antibody resistance [14–20]. Retaining H655Y in every Omicron sub-lineage suggests that the fitness gains associated with recalibrated virus entry outweigh the costs associated with increased susceptibility to antibody neutralization. Overall, our observations reveal a layer of viral adaptation affecting transient S conformations and provide frameworks to assess the properties of SARS-CoV-2 and other pathogenic CoVs.

## Materials and Methods

### Cell lines

HEK293T (a gift from Dr. Ed Campbell, Loyola University Chicago) and Vero-E6-hACE2-hTMPRSS2 (a gift from Dr. Stanley Perlman, University of Iowa) cells were maintained in DMEM-10% FBS [Dulbecco's Modified Eagle Media (DMEM) containing 10 mM HEPES, 100 nM sodium pyruvate, 0.1 mM non-essential amino acids, 100 U/ml penicillin G, and 100 μg/ml streptomycin, and supplemented with 10% fetal bovine serum (FBS, Atlanta Biologicals)]. All cell lines were cultured at 37°C with 5% $CO_2$.

### Plasmid construction

Full-length SARS-CoV-2 S, E, M, and N genes (GenBank: NC_045512.2) were synthesized previously by Genscript, Inc. as human codon-optimized cDNAs, and inserted into pcDNA3.1 expression vectors [10,104]. Omicron BA.1 S was synthesized previously (Intergrated DNA Technologies) [23,24]. All S recombinants were constructed via NEBuilder HiFi DNA Assembly (NEB #E2621S). HiBiT-N was constructed by fusing HiBiT peptide (VSGWRLFKKIS) and linker (GSSGGSSG) coding sequences to the 5' end of the N gene, as described in [10,104]. ΔS-

luc-GFP bacmid (a gift from Dr. Balaji Manicassamy, University of Iowa) was constructed as described in [83,84]. The pCMV-LgBiT expression plasmid was purchased from Promega. pcDNA3.1-hACE2-C9 was obtained from Dr. Michael Farzan, Scripps Florida. pcDNA3.1-hACE2-LgBiT was constructed by fusing the coding sequence of LgBiT to the 3' end of hACE2 gene. pHEF-VSVG-Indiana was constructed previously [64]. 4A8 and CC25.106 antibody expression plasmids were synthesized by using their published VH and VL sequences [43,57] to replace their counterparts in pVITRO1-M80-F2-IgG1/κ (a gift from Andrew Beavil, Addgene plasmid # 50383; http://n2t.net/addgene:50383; RRID:Addgene_50383, [105]). HiBiT-tagged antibodies were constructed by fusing linker (GSSGGSSG) and HiBiT peptide (VSGWRLFKKIS) coding sequences to the 3' end of the antibody CH.

## Virus-like particles (VLPs) and extracellular vesicles (EVs)

HiBiT-N tagged VLPs were produced as described previously [10,23,24,59–62]. Briefly, equimolar amounts of full-length CoV S, E (envelope), M (membrane) and HiBiT-N encoding plasmids were LipoD (SignaGen, cat: SL100668)-transfected into HEK293T cells. To produce spikeless "No S" VLPs, the S expression plasmids were replaced with empty vector plasmids. At 6 h post-transfection, cells were replenished with fresh DMEM-10% FBS. HiBiT-N VLPs were collected in FBS-free DMEM from 24 to 48 h post-transfection. FBS-free DMEM containing HiBiT-N VLPs were clarified by centrifugation (300xg, 4˚C, 10 min, 3000xg, 4˚C, 10 min).

To obtain purified viral particles, clarified VLP-containing FBS-free DMEM was subjected to either size-exclusion chromatography (SEC) or density ultracentrifugation. For SEC, samples were first concentrated 100-fold by ultrafiltration (Amicon, 100 kDa) before SEC (qEV original, Izon, Inc., following manufacturer protocols). VLPs were eluted from columns into 2x FBS-free DMEM. Peak VLP fractions were identified after lysis of VLPs by adding Passive Lysis Buffer (Promega, cat: E1941) and LgBiT and measuring complemented Nluc in a luminometer. For density ultracentrifugation, samples were laid over 20% (wt/wt) sucrose, and spun (SW28, 7500 rpm, 4˚C, 24 h [106]). The resulting pellet was resuspended in FBS-free DMEM. For downstream experiments, VLP inputs were normalized based on their Nluc activity upon adding Passive Lysis Buffer and LgBiT complementation. Samples were stored at -80˚C until use.

hACE2-LgBiT EVs were obtained as described previously [10,23,24,59–62]. Briefly, HEK293T target cells were LipoD-transfected with pcDNA3.1-hACE2-LgBiT. At 6 h post-transfection, transfection media were removed, rinsed, and replaced with FBS-free DMEM. Media were collected at 48 h post-transfection, clarified (300xg, 4˚C, 10 min; 3000xg, 4˚C, 10 min), and concentrated 100-fold by ultrafiltration (Amicon, 100 kDa). EVs were then purified using size-exclusion chromatography (qEV original, Izon, Inc.) using PBS pH 7.4 as eluant. Peak EV fractions were identified by adding HiBiT-containing Passive Lysis Buffer and subsequent Nluc measurement by luminometry. EVs were stored at 4˚C.

## Cell-free fusion assay

Cell-free fusion assays were performed according to established protocols [10,23,24,59–62]. Briefly, at 4˚C, HiBiT-N VLPs and hACE2-LgBiT EVs were mixed with nanoluc substrate (Promega #N2420), DrkBiT (10 μM, peptide sequence VSGWALFKKIS [107], synthesized by Genscript), and trypsin (Sigma #T1426; 1 ng/μl) in 384-well multiwell plates. Sample plates were then loaded into a Glomax luminometer maintained at 37˚C. Nluc accumulations were recorded over time. VLP-EV cell-free fusions were quantified as the fold increase of Nluc signal from S-bearing VLPs over the signal from spikeless (No S) VLP background control.

For S2' cleavage assays, HiBiT-N[+] VLPs bearing indicated recombinant SARS-2-S were incubated with vehicle (PBS), 4A8 (500 nM), or CC25.106 (500 nM) for 30 min at 37˚C. Subsequently, hACE2-LgBiT EVs were added and incubated for 30 min at 37˚C. Lastly, trypsin (1 ng/μl or otherwise indicated) were added and incubated for 30 min at 37˚C. A portion of the resulting mixture was mixed with Nluc substrate and fusion was measured by ways of RLU. SDS-solubilizer was added to the rest of the mixture for Western blot analysis.

For 4A8 or CC25.106 antibody-mediated VLP neutralizations, VLPs were incubated with indicated dilutions of antibody for 30 min at 37˚C before adding hACE2-LgBiT EVs, substrate, DrkBiT, and trypsin. For HR2 peptide neutralization, VLPs were incubated with indicated dilutions of SARS-CoV-2 HR2-cholesterol (gift from Matteo Porotto and Anne Moscona, Columbia University [62,79,80]) for 30 min at 37˚C before adding hACE2-LgBiT EVs, substrate, DrkBiT, and trypsin. For time-of-addition experiments, VLPs, EVs, substrate, DrkBiT, and trypsin were mixed on ice and membrane fusion was initiated by shifting to 37˚C. Then, HR2-cholesterol (100 nM), CC25.106 (300 nM), or hACE2-Fc (63 nM, made previously [10,23,24]) was added after 0, 20, 40, 60, 80, 120, or 180 min. All conditions were carried out to 180 min, and RLU was read.

## VLP-antibody binding assay

Biotinylated VLPs (biotin-VLPs) were obtained as described in [32], with modifications. VLPs were first purified from transfected cell supernatant through 20% sucrose in VLP buffer (50 mM HEPES [pH 7.5], 10 mM $MgCl_2$, 10 mM $CaCl_2$, 0.1% biotin-free BSA [Thermo Scientific, cat: 37525]) via ultracentrifugation (SW28, 25,000 rpm, 4˚C, 3 h). The pelleted VLPs were resuspended in VLP buffer and incubated with 0.08 mg/ml DSPE-PEG2000-biotin (Avanti Polar Lipids, cat: 880129), rotating for 1 h at room temperature. Biotin-VLPs were then purified via ultracentrifugation (SW55, 38,000 rpm, 4˚C, 1 h) through 20% sucrose in VLP buffer. Pelleted biotin-VLPs were resuspended in VLP buffer and stored at -80˚C until use.

VLP-antibody binding assay was done by first mixing biotin-VLPs in VLP buffer with the indicated HiBiT-antibodies (0.2 nM) with or without hACE2 EVs, rotating for 1 hr at 37˚C. Then, Dynabeads MyOne Streptavidin T1 (Invitrogen, cat: 65601) were added and rotated for 1 hr at room temperature. Magnetic pulldown was subsequently performed according to product manual, involving three rinses using wash buffer (25 mM Tris, 150 mM NaCl, pH 7.2, 0.1% biotin-free BSA, 0.01% TWEEN-20). The final pellet was resuspended in Passive Lysis Buffer and antibody retention quantified as Nluc activity via HiBiT-LgBiT complementation. For antibody binding kinetics experiments, biotin-VLPs were incubated with Dynabeads and rotated for 1 h at room temperature prior to the addition of HiBiT-antibody at indicated times.

## ΔS-luc-GFP replicon virions

ΔS-luc-GFP replicon virion stock was made as described in [83,84], with modifications. Briefly, HEK293T cells were LipoD-transfected with both ΔS-luc-GFP and VSVG. At 6 h post-transfection, cells were replenished with fresh DMEM-10% FBS. VSVG[+] ΔS-luc-GFP replicon virions were collected in DMEM-10% FBS from 48 to 72 h post-transfection and clarified by centrifugation (300xg, 4˚C, 10 min; 3000xg, 4˚C, 10 min). To ensure successful downstream incorporation of SARS-CoV-2 S, VSVG[+] ΔS-luc-GFP replicon virions were subjected to three rounds of enrichments: First, 293T cells were transfected with VSVG. 24 h later, VSVG[+] ΔS-luc-GFP replicon virions were inoculated onto the transfected cells. 24 h later, cells were re-transfected with VSVG. 24 h later, the supernatant was collected and clarified by centrifugation (300xg, 4˚C, 10 min; 3000xg, 4˚C, 10 min). After three rounds of enrichment, the resulting VSVG[+] ΔS-luc-GFP replicon virion stocks were aliquoted and stored at -80˚C until use.

To generate SARS-2-S$^+$ $\Delta$S-luc-GFP replicon virions, 293T cells were LipoD-transfected with the indicated recombinant SARS-2-S constructs. At 6 h post-transfection, cells were replenished with fresh DMEM-10% FBS. At 24 h post-transfection, VSVG$^+$ $\Delta$S-luc-GFP replicon virion stock was inoculated onto the transfected cells for 3 h. The cells were then rinsed three times with DMEM-10% FBS and replenished with fresh DMEM-10% FBS for 2 h. Then, the cells were rinsed three times with FBS-free DMEM and replenished with fresh FBS-free DMEM. 24 h later, FBS-free DMEM containing SARS-2-S$^+$ $\Delta$S-luc-GFP replicon virions were collected and clarified by centrifugation (300xg, 4°C, 10 min; 3000xg, 4°C, 10 min).

For $\Delta$S-luc-GFP replicon virion transduction assays, Vero-E6-hACE2-hTMPRSS2 cells were first rinsed once with FBS-free DMEM. Then, $\Delta$S-luc-GFP replicon virions bearing the indicated viral glycoproteins were inoculated for 2 h. Then, cells were rinsed three times with FBS-free DMEM and replenished with fresh DMEM-10% FBS. 24 h later, cells were microscopically assessed for GFP, and supernatant Gluc activity quantified as entry signal by adding substrate (1.1 M NaCl, 2.2 mM Na2EDTA, 0.22 M KH2PO4, 1.3 mM NaN3, .44 mg/mL Bovine Serum Albumin, 2.5 μM Coelenterazine [pH 5]) and luminescence read using a Glomax luminometer. For antibody or HR2 peptide -mediated viral neutralization, replicon particles were incubated with SARS-2-S1B-ab (8F1H7), CC25.106, or HR2-cholesterol at indicated concentrations for 30 min at 37°C before inoculation.

## Western blot and antibodies

Samples in SDS solubilizer [0.0625 M Tris·HCl (pH 6.8), 10% glycerol, 0.01% bromophenol blue, 2% (wt/vol) SDS, +/- 2% 2-mercaptoethanol] were heated at 95°C for 5 min, electrophoresed through 8% (wt/vol) polyacrylamide-SDS gels, transferred to nitrocellulose membranes (Bio-Rad), and incubated with mouse monoclonal anti-SARS-S2 (ThermoFisher, cat: MA5-35946), goat anti-human IgG (sc-2453, Santa Cruz Biotechnologies), mouse anti-C9 (EMD Millipore, cat: MAB5356), rabbit anti-SARS-N (SinoBiological, cat: 40143-R001), or purified LgBiT-substrate cocktail (Promega, cat: N2420). After incubation with appropriate HRP-tagged secondary antibodies and chemiluminescent substrate (Fisher Scientific, cat: PI32106), the blots were imaged and processed with a FlourChem E (Protein Simple).

Recombinant antibodies 4A8, 4A8-HiBiT, CC25.106, and CC25.106-HiBiT were produced by LipoD-transfecting HEK293T cells with expression plasmids as described in [23]. Transfected cells were incubated in FBS-free DMEM containing 2% (wt/vol) Cell Boost 5 (Hyclone). Conditioned media were collected on days 4 and 7, clarified free of debris (300xg, 4°C, 10 min; 4500xg, 4°C, 10 min), and antibodies then purified using HiTrap Protein A High Performance Columns (Cytiva, cat: 29048576) according to the manufacturer instructions. Purified proteins were dialyzed in PBS [pH 7.4], quantified spectrophotometrically and stored at -20°C until use.

## Quantification and statistical analysis

All titration curves are one representative of at least three biological repeats. For these graphs, mean and error are shown based on three technical replicates. To quantitatively compare the effects of spike mutations on S functions, the IC50 or T$_{50}$ values from each biological replicate were pooled, and subsequently normalized to the ancestral D614G S control, whose IC50 or T$_{50}$ values were set to 1 or 0, respectively. Mean and SEM are shown based on data from biological repeats. Values from all conditions were tested for their deviation from the reference value of the ancestral D614G S, and their statistical significances were determined using one-way ANOVA. All graphs and statistical analyses were completed using Prism 8 (GraphPad). *P*-values were indicated in figure legends.

## Supporting information

**S1 Fig. S incorporation onto VLPs.** The incorporation of recombinant S proteins on VLPs. SEC-purified VLPs bearing various recombinant S proteins were subjected to WB analysis of S incorporations. $S_{UNC}$, S2, S2', and HiBiT-N are labeled. S1A or S1B domain-swapped recombinant S proteins affected S incorporation onto VLPs. This is caused by differential S maturation in producer cells, and have been found to not affect their differential sensitivity towards entry inhibitors [24].
(TIF)

**S2 Fig. The characteristics of Omicron Y655H revertant S. A and B.** The SARS-2 VLP cell-free fusion neutralization profile of stem-helix antibody CC25.106 (**A**) or HR2-chol (**B**). Indicated concentrations of inhibitors were incubated with VLPs bearing either Omicron BA.1 or Omicron BA.1 Y655H revertant S for 30 min at 37˚C before EVs, DrkBiT, substrate, and trypsin were added and RLU quantified. Left, dose-response curves between treatment and % fusion. Right, experiment on the left was repeated three times (N = 3) and the doses generating 50% inhibition (IC50) were plotted. Mean and SEM are depicted. Deviations from the reference value of 1.0 (Omicron BA.1 S) were analyzed by one-sample t tests. *, $p < 0.05$; **, $p < 0.01$. C. Cell-free membrane fusion kinetics of VLPs bearing either Omicron BA.1 or Omicron BA.1 Y655H revertant recombinant S proteins. Left, fusion kinetics curves from a representative experiment, where RLU was read every 5 min up to 180 min. Right, experiment on the left was repeated four times (N = 4) and the times generating 50% fusion (T50) were plotted. Mean and SEM are depicted. Deviations from the reference value of 0 (Omicron BA.1 S) were analyzed by one-sample t tests. ***, $p < 0.001$.
(TIF)

**S1 Data. Data underlying Figs 1–5 and S2.**
(XLSX)

## Acknowledgments

We thank Matteo Porotto and Anne Moscona (Columbia University) for the antiviral HR2 peptides used in this study. We thank Balaji Manicassamy (University of Iowa) for providing the ΔS-luc-GFP Replicon.

## Author Contributions

**Conceptualization:** Enya Qing, Tom Gallagher.

**Data curation:** Enya Qing, Julisa Salgado, Alexandria Wilcox.

**Formal analysis:** Enya Qing.

**Funding acquisition:** Tom Gallagher.

**Investigation:** Enya Qing, Julisa Salgado, Alexandria Wilcox, Tom Gallagher.

**Methodology:** Enya Qing, Julisa Salgado, Alexandria Wilcox, Tom Gallagher.

**Project administration:** Enya Qing, Tom Gallagher.

**Resources:** Tom Gallagher.

**Supervision:** Enya Qing, Tom Gallagher.

**Validation:** Enya Qing, Tom Gallagher.

**Visualization:** Enya Qing, Tom Gallagher.

**Writing – original draft:** Enya Qing, Tom Gallagher.

**Writing – review & editing:** Enya Qing, Tom Gallagher.

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
