## [Decision Letter · Decision Letter 0]

17 Sep 2024

Dear dr. Gallagher,

Thank you very much for submitting your manuscript "SARS-CoV-2 Omicron variations reveal mechanisms controlling cell entry dynamics and antibody neutralization" for consideration at PLOS Pathogens. As with all papers reviewed by the journal, your manuscript was reviewed by members of the editorial board and by several independent reviewers. The reviewers appreciated the attention to an important topic. Based on the reviews, we are likely to accept this manuscript for publication, providing that you modify the manuscript according to the review recommendations.

Sincerely,

Bart L. Haagmans

Academic Editor

PLOS Pathogens

Alexander Gorbalenya

Section Editor

PLOS Pathogens

Michael Malim

Editor-in-Chief

PLOS Pathogens

orcid.org/0000-0002-7699-2064

Reviewer Comments (if any, and for reference):

Reviewer's Responses to Questions

**Part I - Summary**

Reviewer #1: The manuscript entitled “SARS-CoV-2 Omicron variations reveal mechanisms controlling cell entry dynamics and antibody neutralization” by Qing and colleagues demonstrated how the H655Y substitution in the Omicron Spike protein alters viral entry kinetics and membrane fusion. Multiple studies have shown the role of the H655Y substitution in stabilizing Spike and preferential cathepsin-mediated entry of Omicron in cell lines. However, Qing and colleagues showed that the H655Y substitution alters viral fusion kinetics, potentially by prolonging the extended intermediate state of S prior to viral and target cell membrane fusion. It is understandable that such a state would increase the chance of antibodies against these epitopes to bind and prevent fusion.

Reviewer #2: The manuscript by Tang and colleagues investigates the impact of the H655Y mutation, frequently found in Omicron S1 subunits, on the membrane fusion process. This mutation delays membrane fusion by affecting a post-S2’ cleavage event, which is analyzed using a neutralizing antibody targeting the stem-helix region of S2. The author proposes a new model of SARS-CoV-2 fusion based on their findings. The experiments are generally well-executed, and the conclusions are supported by the data. That said, the manuscript would benefit from a more detailed discussion of existing studies, including comparisons between cell-cell fusion vs cell-free fusion of D614G and Omicron, and an explanation of the discrepancies in the literature regarding the action of H655Y. Additionally, confirming the results with other Omicron spike proteins would strengthen the findings. It would also be valuable to discuss whether similar mechanisms might apply to other human pathogenic coronaviruses, particularly SARS-CoV.

Reviewer #3: This is an excellent study that presents a modified model of coronavirus fusion activation through study of SARS-Cov-2, in particular the H655Y mutation that is found in Omicron variants. The work nicely illustrates the complexities inherent to the evolution of the viral spike gene and how genomic changes play out in terms of altered function.

**Part II – Major Issues: Key Experiments Required for Acceptance**

Reviewer #1: None

Reviewer #2: • Figures 3F and 3G: Statistical analysis is needed and should be included.

• Figures 5D and 5E: The differences between D614G and Omicron appear minimal in the plotted data despite p-values < 0.001. This discrepancy needs clarification. Additionally, p-values for Figure 5C should be reported.

Reviewer #3: 1) Much of the study hinges on differences between a previously reported neutralizing fusion-inhibitory MAb 4A8 (which recognizes the NTD) and a distinct MAb CC25.106 which recognizes the stem-helix. Using their VLP/EV system (which allows a degree of mechanistic specificity that would be difficult to obtain in cell-based assays), the authors do a good job in characterizing the differences between these two MAbs in terms of their action and the details of how the fusion reaction is inhibited, and link to a specific mutation, H655Y

2) As MAb CC25.106 is key to the present manuscript, but described in detail elsewhere (ref 57), the authors should better introduce this MAb for the current readers – eg epitope localization, cross-reactivity etc

3) Fig 1/2 are pretty narrow in terms of the range of SARS-CoV-2 species that are examined – this may be due to the limitations of assay used, but information of other VOCs, including a range of Omicrons and WA1/B.1 would be useful, especially as H655Y occurs in non-Omicron VOCs (eg Gamma)

4) The Discussion is well written and covers a lot of complex material on SARS-CoV-2 very well

**Part III – Minor Issues: Editorial and Data Presentation Modifications**

Reviewer #1: Although not critical for the paper, the authors should determine the IC50 of CC25.106 against newer Omicron variants, such as JN.1 or KP.3, which have the P681R substitution at the cleavage site, shown to improve fusogenicity.

More importantly though, the authors should show the reversion of the H655Y phenotype in Omicron (mutating the Y back to a H). This is relevant for Figure 2 to show fusion blocking, Figure 4 to show the kinetics of fusion and Figure 5 to show this reversion in the context of live-cell virus entry assays. Does the reversion of this substitution result in even faster entry in the backbone of Omicron?

In Figure 3 it would be interesting to see pre-fusion S binding in the context of Omicron S and Omicron S with the H655Y substitution reverted.

Textually, the choice of “variations” in the title is confusing to me. The authors observed the effect of a single substitution, H655Y on antibody binding and S biochemistry during entry. Also in the abstract, what do the authors mean by “Omicron variations in the peripheral “S1” domains altered late-stage fusion dynamics”. They could just say instead that “peripheral S1 domains specific to Omicron altered late-stage fusion dynamics” or just focus on the single substitution of interest.

Figure 1, I suggest to split panel 1B into 1B and 1C.

To better follow the different domains of Spike and how they are affected, a diagram showing the Spike protein and all of its domains would be useful.

Like in Figure 2D, similarly to showing at least whether reverting the H655Y substitution in the Omicron backbone would revert the phenotype, it would be interesting if the authors would look at all the substitutions added to the D614G backbone instead reverted in the Omicron backbone.

Please quantify the proteolytic cleavage in Figure 3C.

Reviewer #2: • Omicron Lineages: Since Omicron has evolved into multiple lineages, the manuscript should specify the BA.1 lineage.

• Terminology: The term "CoV" is used in many places to refer to SARS-CoV-2 and should be corrected for clarity.

Reviewer #3: Minor points, some of the wording in the text is not the best (eg “world dominance”, and what is meant by “opposition” in the abstract), are SD1 & 2 better defined as domains C & D?

PLOS authors have the option to publish the peer review history of their article (what does this mean?). If published, this will include your full peer review and any attached files.

Reviewer #1: No

Reviewer #2: No

Reviewer #3: No

Figure Files:

Data Requirements:

Reproducibility:

References:

---

## [Editor Report · Decision Letter 1]

17 Nov 2024

Dear dr. Gallagher,

We are pleased to inform you that your manuscript 'SARS-CoV-2 Omicron variations reveal mechanisms controlling cell entry dynamics and antibody neutralization' has been provisionally accepted for publication in PLOS Pathogens.

Best regards,

Bart L. Haagmans

Academic Editor

PLOS Pathogens

Alexander Gorbalenya

Section Editor

PLOS Pathogens

Michael Malim

Editor-in-Chief

PLOS Pathogens

orcid.org/0000-0002-7699-2064
---

## [Editor Report · Acceptance letter]

23 Nov 2024

Dear Gallagher,

We are delighted to inform you that your manuscript, "SARS-CoV-2 Omicron variations reveal mechanisms controlling cell entry dynamics and antibody neutralization," has been formally accepted for publication in PLOS Pathogens.

Best regards,

Michael Malim

Editor-in-Chief

PLOS Pathogens

orcid.org/0000-0002-7699-2064